# Design of a Variable Stiffness Gecko-Inspired Foot and Adhesion Performance Test on Flexible Surface

**DOI:** 10.3390/biomimetics7030125

**Published:** 2022-09-05

**Authors:** Zhiwei Yu, Jiahui Fu, Yu Ji, Benhua Zhao, Aihong Ji

**Affiliations:** 1College of Mechanical and Electrical Engineering, Nanjing University of Aeronautics and Astronautics, Nanjing 210016, China; 2Mechanical Products Division, Beijing Spacecrafts, Beijing 100094, China

**Keywords:** gecko-inspired robot, dry adhesion foot, variable stiffness design, flexible surface

## Abstract

Adhesion robots have broad application prospects in the field of spacecraft inspection, repair, and maintenance, but the stable adhesion and climbing on the flexible surface covering the spacecraft has not been achieved. The flexible surface is easily deformed when subjected to external force, which makes it difficult to ensure a sufficient contact area and then detach from it. To achieve stable attachment and easy detachment on the flexible surface under microgravity, an adhesion model is established based on the applied adhesive material, and the relationship between peeling force and the rigidity of the base material, peeling angle, and working surface stiffness is obtained. Combined with the characteristics of variable stiffness structure, the adhesion and detachment force of the foot is asymmetric. Inspired by the adhesion-detachment mechanism of the foot of the gecko, an active adhesion-detachment control compliant mechanism is designed to achieve the stable attachment and safe detachment of the foot on the flexible surface and to adapt to surfaces with different rigidity. The experimental results indicate that a maximum normal adhesion force of 7.66 N can be generated when fully extended, and the safe detachment is achieved without external force on a flexible surface. Finally, an air floating platform is used to build a microgravity environment, and the crawling experiment of a gecko-inspired robot on a flexible surface under microgravity is completed. The experimental results show that the gecko-inspired foot with variable stiffness can satisfy the requirements of stable crawling on flexible surfaces.

## 1. Introduction

Spacecraft is a typical high value and complex system [1]. Timely detection and maintenance are an important part of ensuring its working ability and service life. According to the needs of thermal protection, the surface of the spacecraft is covered by flexible protection materials. Ensuring the stable adhesion of the robot on the surface of the flexible material is a difficulty in space engineering. At present, the attachment methods include magnetic adsorption [2,3], electrostatic adsorption [4,5], vacuum adsorption [6,7], and dry adhesion [8,9,10,11,12,13,14,15]. These adhesion methods have their own characteristics. Magnetic adsorption requires the working surface to be ferromagnetic; electrostatic adsorption requires a high voltage to maintain the adsorption force; vacuum adsorption is not suitable for vacuum environment; dry adhesion is based on Van der Waals force between solid molecules, which is currently known to be the most suitable for space environment.

In 2000, AUTUMN et al. [16] revealed the adhesion characteristics and mechanism of the gecko’s feet. In the past 20 years, the research on dry adhesion climbing robots has made great progress. The dry adhesion climbing robot Stickybot [17] developed by Kim et al. could climb 4 cm/s on a vertical rigid surface. The feet with various anisotropic adhesion material is designed with a pull-up type (Figure 1a). Stickybot-III developed by Elliot et al. [18] used the foot design of the gecko to achieve the uniform distribution of loads on the toe pads of the feet and reduce the overturning force. However, passive attachment cannot regulate or reduce the adhesion force (Figure 1b). The Abigail series of dry adhesives hexagonal climbing robots were developed by Menon et al. [19,20,21,22,23]. Abigaille adopted passive striking method, and it can bend its ankles (Figure 1c); Abigaille-II also used it. As shown in Figure 1d, for the toe pad of foot, the dry adhesive material is attached to a sheet attached to the outside of the foot, which can obtain a larger adhesive area and reduce adhesion. Abigaille-III, shown in Figure 1e, used an active desorption method, which uses a cam mechanism to disengage the toe pad of the foot. This desorption method is only suitable for rigid surfaces.

Compared with the adhesion and desorption of the rigid surface, the flexible surface is more difficult to adhere stably. The reasons are: (a) The flexible surface deforms greatly under force, and it will be difficult to fit the foot; (b) the surface is prone to vibrate under the action of external force, which may cause robotic resonance and lead to unstable exercise. Under microgravity, the adhesion of flexible surface needs to have the characteristics of robust adhesion and easy detachment, so the toe pad of the foot needs to adapt actively to control the adhesion.

This article takes the gecko adhesion and desorption mechanism as the starting point. The simulation model of its adhesive force for the adhesive material used is established, and the relationship between desorption force, desorption angle, and the substrate stiffness is analyzed. An active bionic foot with variable stiffness is designed.

## 2. Model and Design

### 2.1. Theoretical Adhesion Model Establishment

Currently, the model that is useful to describe the desorption process is the Kendall Tape model, proposed by Ravin and improved by Kendall [24,25]. Based on the idea of conservation of energy, it is widely applicable to explain the process of elastic thin film from the rigid surface. The adhesion material considered by the Kendall Tape model is the desorption of the thin film material, and the shape of the peel zone and the bending stiffness of the adhesive material are not considered. It does not conform to the research of this paper. Based on the observation of the setal system of the gecko, Pesika proposed the PZ model [26,27], which obtains the relationship between the desorption force and angle of desorption. Since the micropillar on the adhesive material is quite different from the gecko setae in terms of adhesion force, elastic modulus, and stretchable length, the shape of the peel zone may not be consistent with the PZ model. However, the adhesion model proposed in this paper is based on the dry adhesion materials used, which is suitable for this study.

The dry adhesion material used in this paper is PVS (Polyvinyl Siloxane) material. Its micro-morphology is shown in Figure 2a. The surface of the material is neatly arranged with many mushroom head-shaped micropillars. Figure 2b shows the composition of adhesion material, including the micropillars, the base, and the sandwiched layer to stick them together.

The base material is divided into N equivalent base units. The base unit and the micropillar are connected by the rotating joint $J_{1}$. Changing the rotation stiffness of the rotating joint $J_{1}$ is equivalent to changing the bending stiffness of the base material. One end of the adhesion units is hinged to the center of the base unit by the rotating joint $J_{2}$, and the other end is hinged to the working surface by the rotation joint $J_{3}$. In addition, because the working surface is made of flexible material, it is divided into multiple surface units. The adjacent unit is hinged by the rotating joint $J_{4}$. By changing the value of the joint rotation stiffness, it can change the bending stiffness of the working surface equivalently. Let the rotation stiffness of the rotation joint $J_{1}$ be k, the rotation stiffness of the joints $J_{2}$ be $k_{s}$, the length of the base unit be $L_{s}$, the stretch elastic modulus of the adhesion unit be E, and the original length be $l_{AO}$. The cross-section is a circle with radius r. Usually, the length of the adhesion unit is greater than the diameter, and the curvature deformation on it is smaller. Therefore, the torque generated by the adhesive unit is ignored, with only the adhesion and elasticity in the direction of the adhesive unit. The base unit and adhesion unit are numbered from left to right (1, 2, …, n), and Figure 3 is the force analysis of the base unit and adhesion unit in the adhesive model. $\beta_{i}$ is the angle between the base unit and the horizontal direction, $\alpha_{i}$ is the angle between the attachment unit and the connection unit of the base unit, and $\theta_{i}$ is the angle between the adhesive unit and the working surface. The base unit i is also subjected to the tension $f_{Li}$, $f_{Ri}$, torque $M_{Li}$, $M_{Ri}$, and the force $f_{Ai}$ of adhesion units. $\gamma_{Li}$ and $\gamma_{Ri}$ are the angle between $f_{Li}$ and $f_{Ri}$ in the horizontal direction, respectively.

When the model is in a balanced state, there are
(1)$$f_{Li}\cdot \cos\gamma_{Li}+f_{Li}\cdot \cos\left(\alpha_{i}-\beta_{i}\right)=f_{Ri}\cdot \cos\gamma_{Ri}$$
(2)$$-f_{Li}\cdot \sin\gamma_{Li}+f_{Ri}\cdot \sin\gamma_{Ri}=f_{Ai}\cdot \sin\left(\alpha_{i}-\beta_{i}\right)$$
(3)$$M_{Li}-\frac{1}{2}\cdot f_{Li}\cdot \sin\left(\gamma_{Li}-\beta \right)\cdot L_{s}=M_{Ri}+\frac{1}{2}\cdot f_{Ri}\cdot \sin\left(\gamma_{Ri}-\beta_{i}\right)\cdot L_{s}$$
(4)$$f_{Ri-1}=-f_{Li},\;\;M_{Ri-1}=-M_{Li},\;\;\gamma_{Ri-1}=\gamma_{Li}$$

The adhesion unit only has adhesiveness and elasticity. Both are equal in magnitude and opposite in direction. Therefore, the adhesion force generated by a single adhesion unit can be calculated through the following formula.
(5)$$f_{Ai}=\frac{E\cdot S\cdot \left(l_{Ai}-l_{AO}\right)}{l_{AO}}$$

Among them, S is the cross-sectional area of the adhesion unit and $l_{Ai}$ is the length of the adhesive unit at this time. The critical condition of the model is to reach the maximum length $L_{Amax}$ of an adhesion unit. At this time, the adhesion force on the adhesion unit is $f_{Amax}$, assuming that the maximum length of all adhesive units is consistent.

### 2.2. Model Simulation

We established simulation models in Matlab/Multibody. Based on the observation of adhesive material, the base unit length is 100 μm and the thickness is 10 μm. The length of the adhesion unit $l_{AO}$ is 200 μm, the radius r is 20 μm, the maximum length $L_{Amax}$ is 400 μm, and the elastic modulus E is 100 MPa. In addition, the gravity and mass are ignored in the model, but the acceleration will be infinite when the external force is on it for an object without mass. Therefore, a certain damping is given to each joint to ensure the stability of the simulation. Figure 4 is the simulation model established in Multibody. In order to preliminarily explore the relationship between desorption force and desorption angle, the relationship between desorption force and bending stiffness of substrate, and the relationship between desorption force and working surface stiffness, the setting model contains 20 surface units, 20 adhesion units, and 25 base units. The additional five base units are used to simulate some of the adhesion material that has been adhered to. In order to facilitate the adjustment of the perspective and base rigidity of the adhesion, the parameter setting function is added. At the beginning of the simulation, the external force will slowly increase with time and pull up the adhesive material. When the system satisfies the condition of the critical resilience, that is, the length of the adhesion unit reaches $L_{max}$, the simulation ends, and the peeling force is recorded. The simulation experiment contains three variables: the desorption angle θ, the bending stiffness of the base *k* and the bending stiffness of the working surface $k_{s}$. The angles of desorption are 5°, 15°, 30°, 45°, 60°, 75°, 90°, 105°, 120°, 135°, 150°, and 165°. In order to obtain the relationship among the base rigidity, the working surface stiffness, and the force of desorption, the base rigidity in the model can be divided into three types: small, medium, and large. The corresponding *k* values are 10 μN·m/°, 50 μN·m/°, and 250 μN·m/°. The working surface stiffness in the model also includes three types. The bending stiffness $k_{s}$ of the rigid surface is 1000 μN·m/°, the bending stiffness $k_{s}$ of the flexible surface is 2 μN·m/°, and the bending stiffness $k_{s}$ of the super flexible surface is 0.1 μN·m/°. Figure 5, Figure 6 and Figure 7 show the desorption force distributions of rigid, flexible, and super flexible surfaces, respectively. Each figure is made up of arrows in three different colors and a polyline connecting the ends of the arrows. Different colors indicate different base bending stiffness; different arrows in the same color represent the experimental results of different desorption angles. From left to right, the arrows represent the experimental results of desorption angles of 5°, 15°, 30°, 45°, 60°, 75°, 90°, 105°, 120°, 135°, 150°, and 165°. The coordinate value at the end of the arrow indicates the tangential and normal desorption forces.

The following conclusions can be obtained by the simulation results of the adhesive model: (1) The desorption force of adhesive material decreases with the increase of desorption angle, showing asymmetry; (2) the greater the bending stiffness of the base, the greater the desorption force of the adhesive material; (3) the smaller the bending stiffness of the working surface, the smaller the adhesive force can be generated, and the more difficult it is to achieve stable adhesion.

### 2.3. Adhesive Test Experiment

Simulation models are now verified by the desorption experiment of adhesion material. The experimental platform is shown in Figure 8, including a two-dimensional (2D) mobile platform, two-dimensional (2D) force measuring module, fixture, adhesive material, and working surface. The fixture is fixed on the two-dimensional (2D) force measuring module. The fixture and the adhesive material are connected by a 40 mm long adhesive tape. The tape material is very flexible and thin, which can only pass the tension along the tape. The adhesive material completely adheres to the working surface. The size of adhesive material used in the experiment is 30 mm×10 mm, which is completely fitted with the base. The base material is polyvinyl chloride (PVC), and the size is 30 mm×10 mm. In order to verify the effects of different rigidities on the dismissal, three bases with three thicknesses are selected, and the thickness d is 0.03 mm, 0.1 mm, and 0.35 mm, respectively.

In order to verify the impact of perspective on dismissal and ensure the stability of strife and the consistency of perspective, the description was completed by controlling the 2D mobile platform to pull the adhesive material along different angles. During the experiment, 10 groups of different desorption angles were taken under the same substrate stiffness, and the desorption angles were 15°, 30°, 45°, 60°, 75°, 90°, 105°, 120°, 135°, and 150°, respectively. In the initial state, the adhesive material was completely fitted with the working surface. The platform mobile tightened the tape to drive the adhesion materials to be attached. We recorded the descriptions of different base stiffness and different aid angles by using vector diagrams, as shown in Figure 9.

Under the three stiffness, the desorption force shows obvious asymmetry: when θ is large, the adhesion with smaller θ is far greater than that with larger θ. In addition, when d=0.35 mm, the desorption force is greater than d=0.03 mm. When θ=165°, the desorption force of d=0.03 mm is 57.14% of d=0.35 mm. The desorption experiment verifies the correctness of the adhesive model.

### 2.4. Gecko-Inspired Foot Design

Geckos have excellent climbing ability and can achieve stable adhesion movements in horizontal, vertical, and even inverted surfaces. The gecko adhesion process is accompanied by inward grasping of the toes, and the detachment process is accompanied by outward turning of the toes to achieve active detachment (Figure 10b). This behavior ensures that when gecko is attached, its toes uncurl to obtain a larger contact area, while when the gecko is detached, its toes roll up and detach from the flap one by one, ensuring that this process is accompanied by a low and smooth detachment force. The application of this desorption mechanism to the gecko-inspired robot foot can effectively improve the stability of adhesion.

Based on the results of the adhesion model, a gecko-inspired foot with variable stiffness characteristics was designed, as shown in Figure 11. The weight was 243 g, the height was 80 mm, and the width at the widest point was 200 mm.

The structure of the foot is shown in Figure 12. The top layer of the foot is a driving servo and a winding helm disk. The bottom layer is four toes, and each toe consists of a telescopic rod, variable stiffness materials, and adhesion materials. The expansion and rolling of the toe are controlled by the servo relaxing or tightening the traction wire to change the length of the telescopic rod. The telescopic rod is equipped with a spring, which is in an extended state when it is not subjected to external force. At this time, the toe is expanded, resulting in a larger attachment area. The traction line is tightened to drive the telescopic rod to retract, and the toe rolls up to complete the desorption. Top and bottom are connected by ball joints, which provide three passive degrees of freedom, and when the working plane tilts or shakes, the stability of the body will not be affected. At the same time, when the fuselage is slightly inclined, it will not affect the adhesion of the palm on the working surface.

In order to prevent the robot’s feet from shaking when swinging and make the feet parallel to the working surface when touching the working surface, four reset springs were installed on the feet to improve the stability of the structure.

The toe of the telescopic rod was placed in the spring and the wire. When the wire is relaxed, the telescopic rod is in an extended state, and the toe pad of the foot is opened. At this time, it was attached to the working face, so that a larger attachment area could be obtained. When absence is required, the servo drives the rudder to rotate, the traction cable tightens, the telescopic rod contracts, and at the same time drives itself to rise, so that the variable stiffness toe naturally rolls up. The rolling process makes the adhesive material separate from the absence surface, and several absence states in the absence process are shown in Figure 13a.

As shown in Figure 13b, the base material was made of a silicone outer layer wrapped around the inner layer of the metal sheet, and its variable stiffness characteristic was determined by the cross-sectional shape of the sheet. The sheet had different cross-sectional bending coefficients when straightened and bent.

When the metal sheet is bent, its cross-section is approximately as a rectangle, as shown in Figure 14a.

The width of the rectangle is w, the height is h, and its cross-sectional bending coefficient $W_{b}$ is
(6)$$W_{b}=\frac{1}{6}\cdot w^{2}\cdot h$$

When the metal sheet is straightened, the cross-sectional shape is approximately a partial arc, as shown in Figure 14b. The arc angle is $\theta_{s}$, the radius is $r_{s}$, and the thickness is $b_{s}$. Since the thickness and cross-sectional area of the metal sheet remain unchanged when bending and straightening, there is the following relationship:(7)$$b_{s}=h$$
(8)$$r_{s}\cdot \theta_{s}=w$$
(9)$$W_{s}=\frac{w^{2}\cdot b_{s}\cdot \left(2\alpha_{s}+\sin2\alpha_{s}\right)}{8\alpha_{s}\left(\alpha_{s}-\sin\alpha_{s}\right)}$$
where $W_{s}$ is the bending coefficient of the section when the metal sheet is straightened, $\alpha_{s}=0.5\theta_{s}$. From Equations (6)–(9),
(10)$$r_{w}\left(\alpha_{s}\right)=\frac{w_{s}}{w_{b}}=\frac{3\left(2\alpha_{s}+\sin2\alpha_{s}\right)}{4\left(\alpha_{s}-\sin\alpha_{s}\right)}$$
where $r_{w}\left(\alpha_{s}\right)$ is the ratio of the anti-bending coefficient of the cross-section of the mental sheet when it is straight and bent, where $\alpha_{s}\approx 19{}^{\circ}$ is substituted into Equation (10):(11)$$r_{w}\left(19^{{}^{\circ}}\right)\approx 158.70$$

In summary, the bending coefficient of the section of the metal sheet is much greater when it is stretched than when it is bent. When this material is used as the base part to the gecko-inspired foot, the toe root will have a larger bending stiffness and the end will have a smaller bending stiffness.

### 2.5. Adhesion Analysis of the Gecko-Inspired Foot

The adhesion and desorption force are essentially the forces required to detach the feet from each other after they are attached to the working surface. However, the adhesion force of the feet with variable stiffness during adhesion and the detachment force required during detachment show some asymmetry. That is, when the adhesion force of the foot can be firmly attached to the working surface without falling off, and when it needs to be detached, only a small force is required to realize the detachment from the surface. This property comes from the design of the foot. The base of the foot root is in a straight state, while the end is in a rolled-up state. The stiffness of the base in the two states is different, and the corresponding detachment conditions are also different: the base of the root is in a state of large stiffness base detachment, and the end is in a small stiffness detachment state. In addition, when the foot is in the adhesion state, the contralateral toe forms a “Y” structure [28], and the normal detachment force F required to detach it from the working surface is distributed to the four toes. The force state of each toe is close to that in the adhesion model, and the force relationship is shown in Figure 15.

Because the base is not completely rigid when it is straightened, the section near the root must bend to a certain extent, and then the external force will be transferred to the base of the adhesion section along this section. At this time, the required force $F_{p1}$ is larger for the detachment condition of the toe base and the attached material with large rigidity and small angle. When detaching, the required detachment force is generated by the force $F_{p2}$ rolled up by the base. The detachment condition is that the stiffness of the base is small, the detachment angle is large, and the required detachment force is small.

## 3. Results

### 3.1. The Performance Test of the Gecko-Inspired Foot

The stable adhesion between the foot and the working surface in the space environment mainly depends on the normal adhesion force. However, the flexible surface of spacecraft is generally a thin film made of polyimide and metal materials, and the normal contact force has a great influence on the deformation of the flexible surface. Based on the above considerations, the performance test of the adherent foot is carried out for the normal detachment force on the flexible surface. The experimental platform adopts the spacecraft surface film as the flexible surface, which is fixed to the base of the test platform. The toe pad of the foot takes its own weight (73.5 g) as the pre-pressure and the flexible surface, and the area of the material attached to the toe pad is $47.2\;{\mathrm{cm}}^{2}$. Figure 16a shows that the toe is fully extended, the lead rope is fixed to the center of the foot, and then the top of the platform is moved up to pull the foot to complete the desorption. The desorption force during the desorption process is recorded. Figure 16b shows the desorption of the toe pad on the flexible surface in the way of lateral curling, and the desorption force during the process of desorption is recorded. The test results are shown in Figure 16c. When the toe pad of the foot is desorbed by pulling up, the maximum desorption force required is 7.66 N, while, when the toe pad of the foot is desorbed by rolling up, it can be desorbed almost without additional desorption force.

### 3.2. Application of the Gecko-Inspired Feet

In the application of adhesive motion, the foot design of gecko toe pad is combined with a climbing robot to form a bionic gecko robot, as shown in Figure 17. In addition, the test experiments of various movement modes of gecko-inspired robots are carried out in microgravity.

The gecko-inspired robot is fixed on the air floating table bracket, and the continuous spraying at the bottom of the bracket offsets the weight of the bracket itself and the robot, creating a microgravity condition. Combined with the smooth steel plate plane, it eliminates the friction pair between the bottom and the moving ground during the movement. Figure 18a,b shows the linear crawling motion of the gecko-inspired robot on the inverted surface of the polyimide flexible film in a microgravity environment from different perspectives (Appendix A). The experimental results show that the gecko-inspired robot can crawl stably on a flexible surface in microgravity with the help of gecko-inspired toe pads.

## 4. Conclusions

In this paper, an imitation gecko foot is proposed and its adhesion performance is tested. The foot paw adopts an active adhesion-debonding mechanism combining materials with variable bending stiffness to achieve asymmetric adhesion-debonding characteristics.

First, the adhesion model is established based on the microstructure of the dry adhesion PVS material. Through simulation analysis and design of the adhesion experiment, the relationship between adhesion force and adhesion angle, bending stiffness of the substrate and bending stiffness of the working surface was investigated. The results show that the smaller the debonding angle, the greater the bending stiffness of the matrix and the bending stiffness of the working face, and the greater the debonding force required for debonding.

A bionic gecko foot with variable stiffness was then designed according to the results of the adhesive model and the structure with variable bending stiffness. Inspired by the adhesion-desorption behavior of the large gecko paw, an active desorption mechanism was designed to make the paw exhibit asymmetric adhesion-desorption characteristics.

Finally, the adhesion-desorption performance of the bionic gecko paws on flexible surfaces was tested. The results show that the variable stiffness paw design with active desorption mechanism can adapt to the flexible surfaces. The maximum normal adhesion force is 7.66 N, which can be desorbed without additional tension, showing a high degree of asymmetric adhesion-desorption characteristics, laying a foundation for the application of bionic gecko robots on flexible surfaces.

## Figures and Tables

**Figure 1 biomimetics-07-00125-f001:**
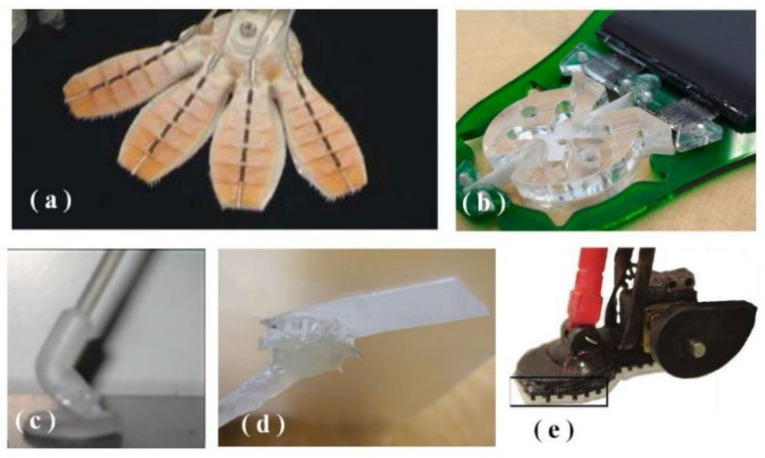
Several representative dry sticking wall robotic feet designs: (**a**) The foot of Stickybot; (**b**) the foot of Stickybot-III; (**c**) the foot of Abigaille; (**d**) the foot of Abigaille-II; (**e**) the foot of Abigaille-III.

**Figure 2 biomimetics-07-00125-f002:**
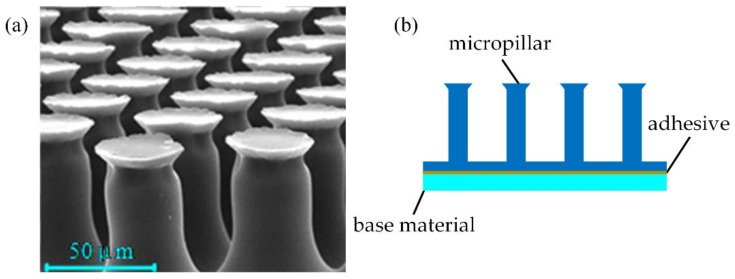
PVS material micro-structure and adhesion material composition: (**a**) The dry adhesion material’s micro-structure; (**b**) the composition of adhesion material.

**Figure 3 biomimetics-07-00125-f003:**
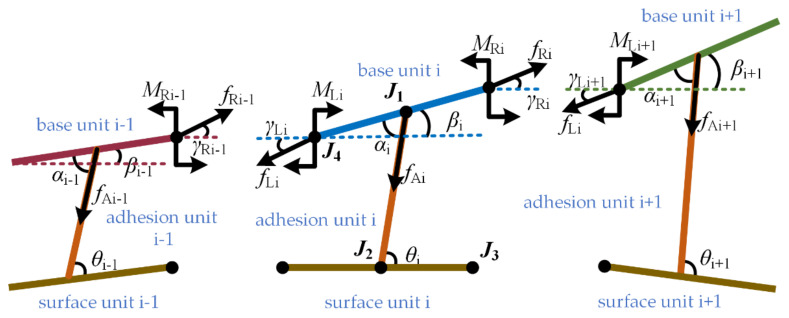
Micro adhesion model of adhesion material.

**Figure 4 biomimetics-07-00125-f004:**
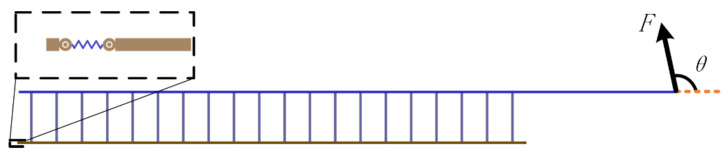
The structure diagram of the simulated model.

**Figure 5 biomimetics-07-00125-f005:**
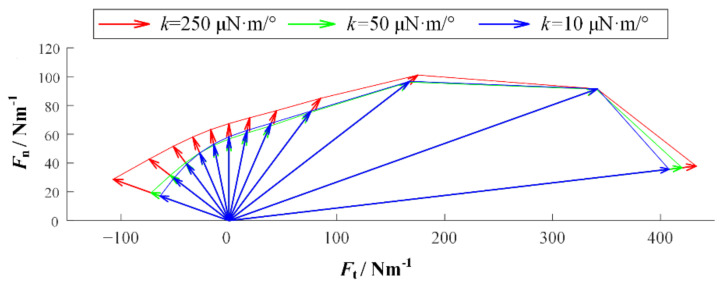
Distribution of desorption forces on rigid surfaces.

**Figure 6 biomimetics-07-00125-f006:**
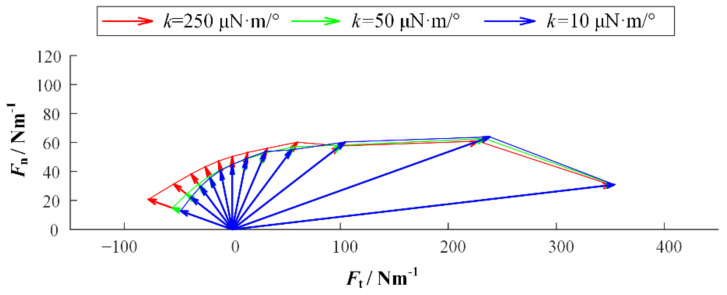
Distribution of desorption forces on flexible surfaces.

**Figure 7 biomimetics-07-00125-f007:**
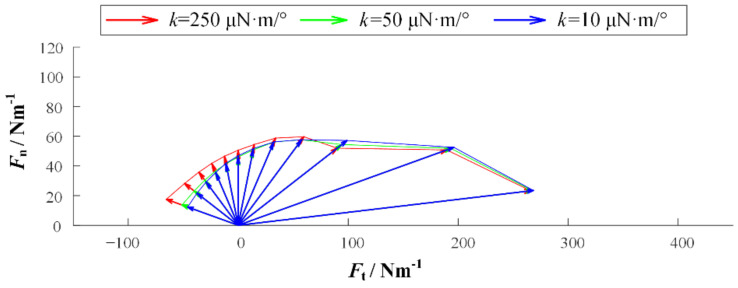
Distribution of desorption forces on super flexible surfaces.

**Figure 8 biomimetics-07-00125-f008:**
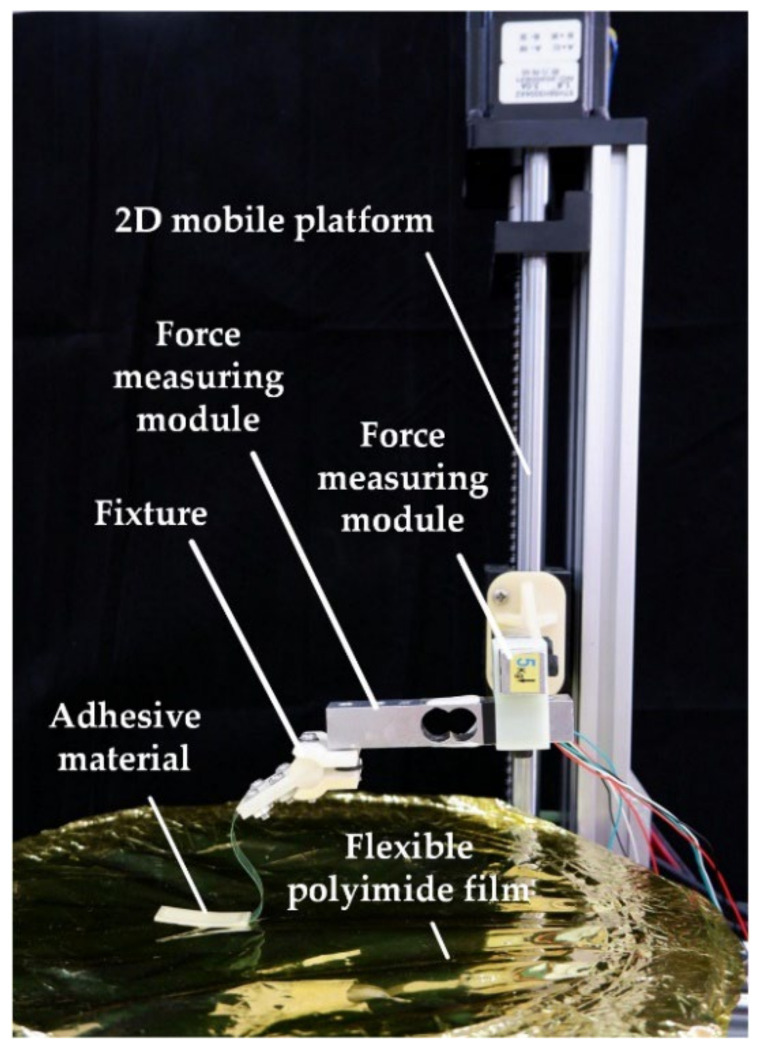
Desorption experimental platform.

**Figure 9 biomimetics-07-00125-f009:**
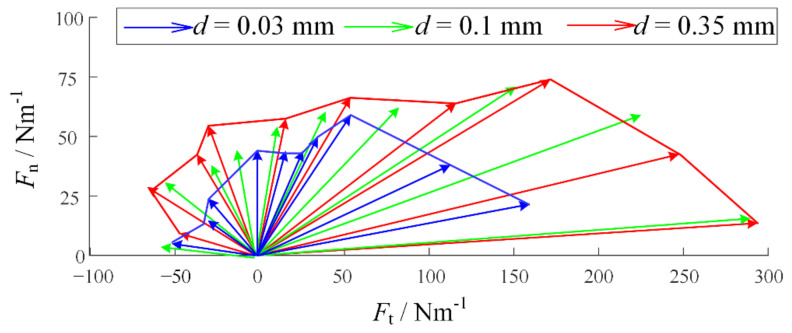
Disposal distribution at different beaded angles.

**Figure 10 biomimetics-07-00125-f010:**
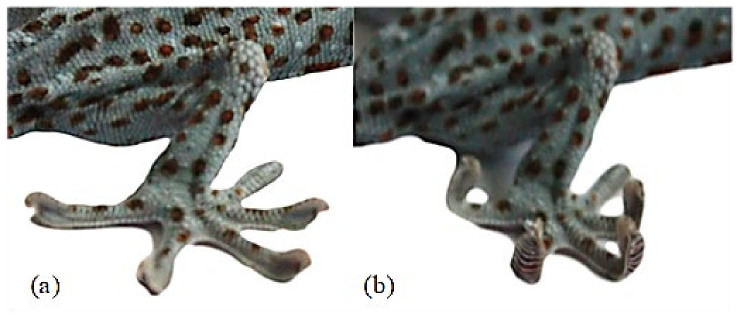
The desorption method of a gecko foot: (**a**) The gecko’s foot is spread out; (**b**) the gecko’s foot is curled up.

**Figure 11 biomimetics-07-00125-f011:**
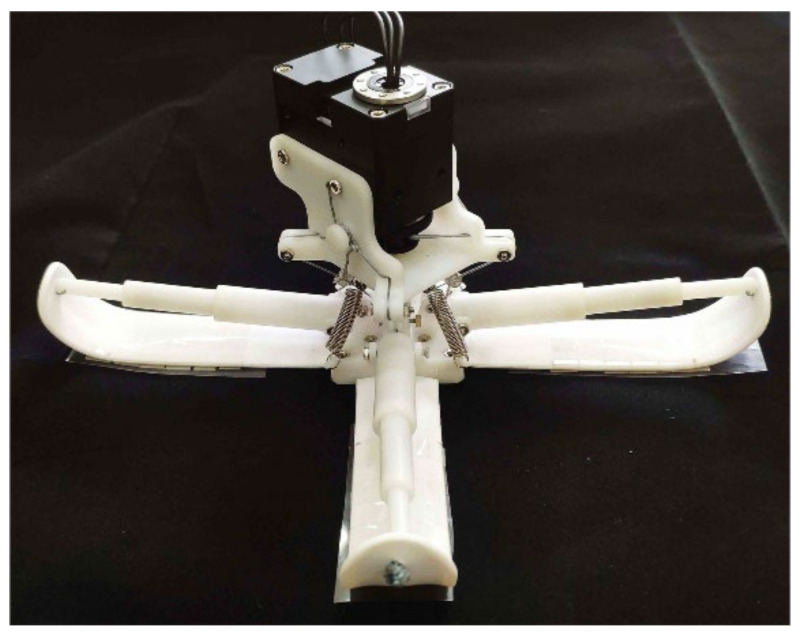
Gecko-inspired foot.

**Figure 12 biomimetics-07-00125-f012:**
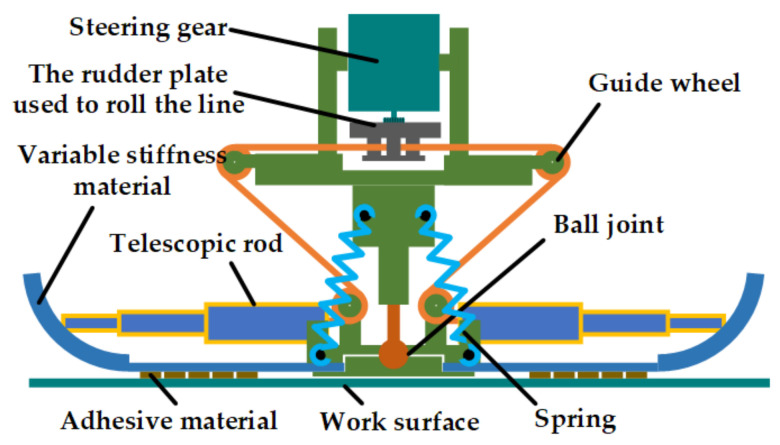
Structure of the gecko-inspired foot.

**Figure 13 biomimetics-07-00125-f013:**
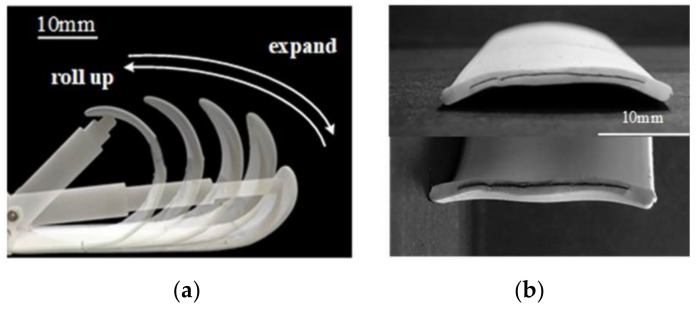
Several states of the toe movement process and substrate material. (**a**) Several states of the toe movement process; (**b**) cross-sectional view of substrate material.

**Figure 14 biomimetics-07-00125-f014:**
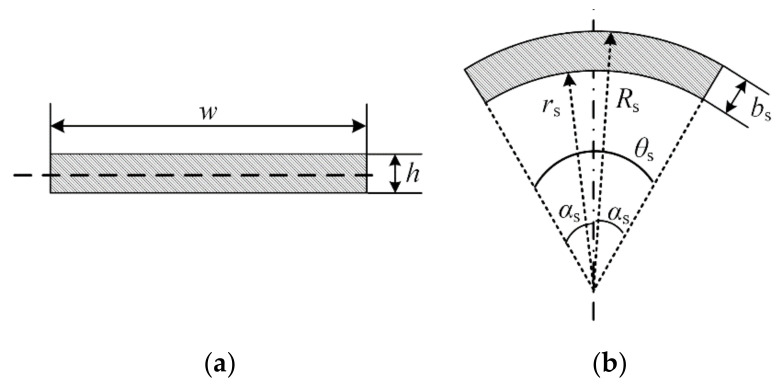
Calculation of section flexural coefficient diagram: (**a**) The metal sheet’s cross-section when it is bent; (**b**) The metal sheet’s cross-section when it is straightened.

**Figure 15 biomimetics-07-00125-f015:**
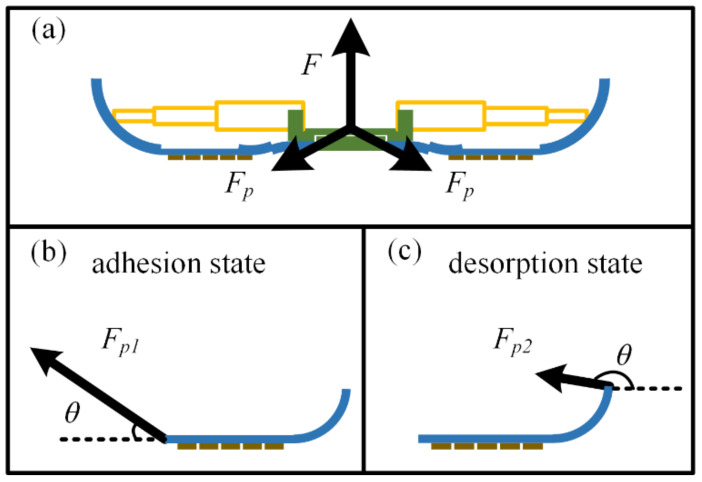
Toe force in different states: (**a**) When the foot is in the adhesion state, the contralateral toe forms a “Y” structure; (**b**) the force state of each toe when the foot is in the adhesion state; (**c**) the force state of each toe when the foot is in the desorption state.

**Figure 16 biomimetics-07-00125-f016:**
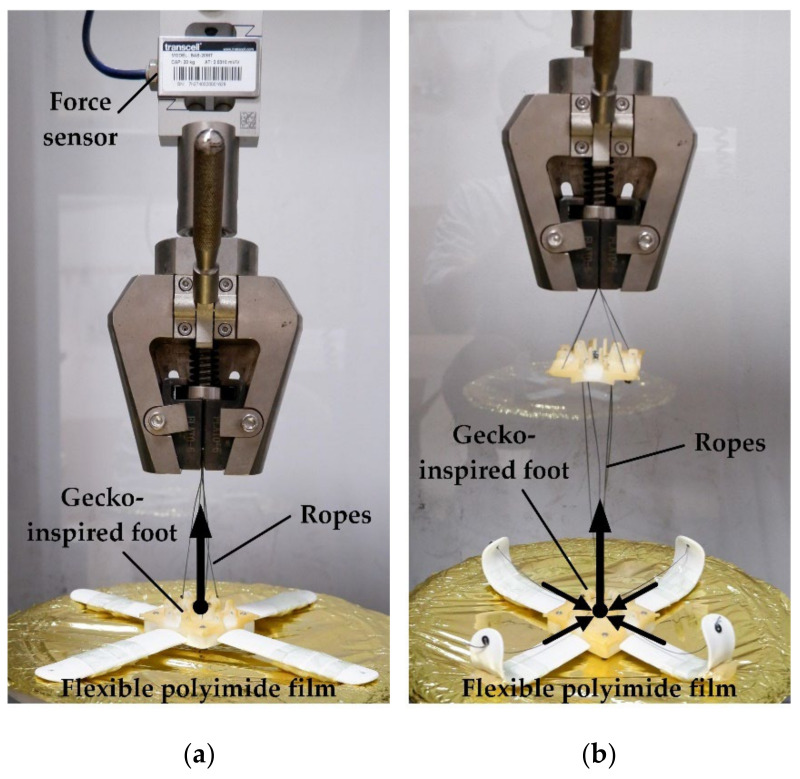
Foot design of the toe pad of gecko feet performance test: (**a**) The toe pad of the foot is desorbed by pulling up; (**b**) the toe pad of the foot is desorbed by rolling up; (**c**) the test results.

**Figure 17 biomimetics-07-00125-f017:**
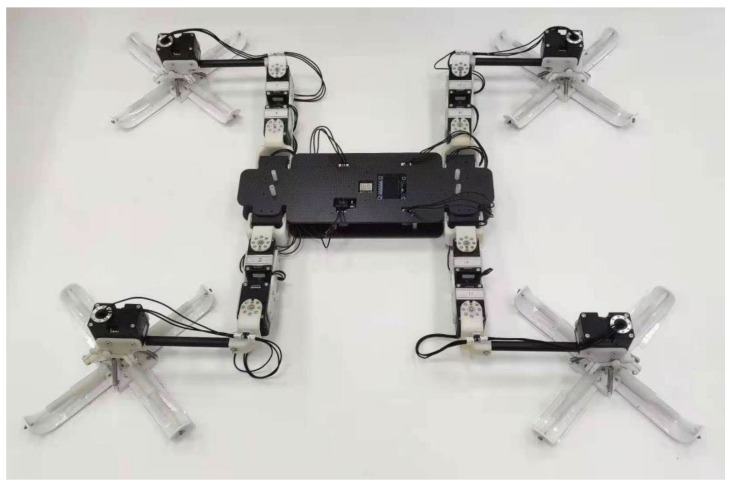
Image of the gecko-inspired robot.

**Figure 18 biomimetics-07-00125-f018:**
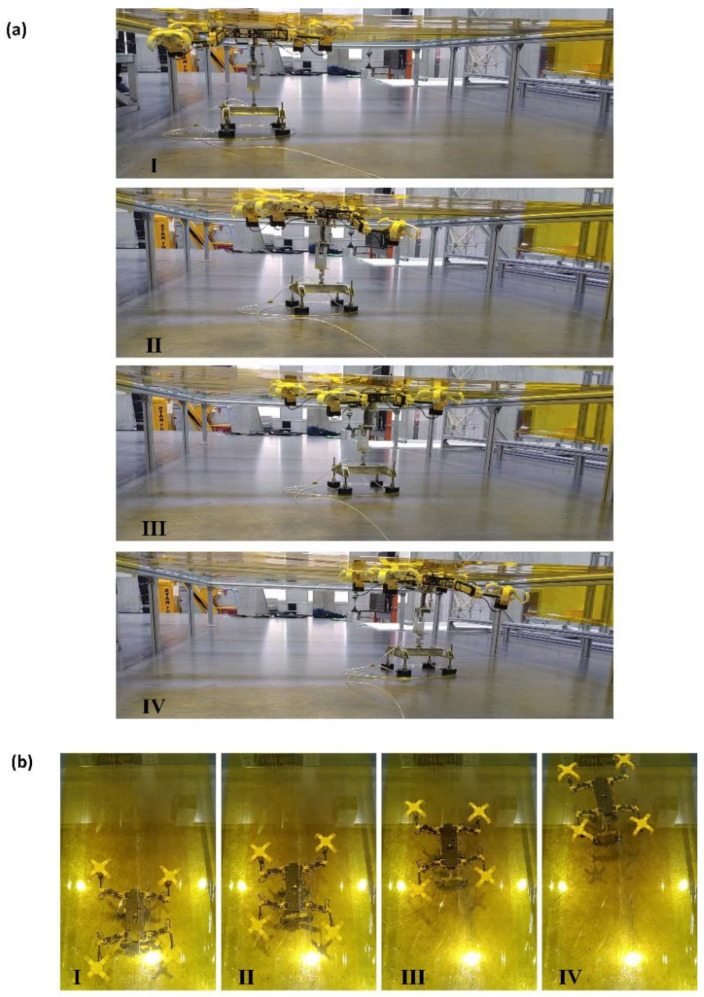
Experiment of gecko-inspired robot crawling on flexible surface: (**a**) Adhesion climbing experiment of gecko-inspired robot on flexible surface by using air floating platform (side view); (**b**) adhesion climbing experiment of gecko-inspired robot on flexible surface by using air floating platform (top view).

## Data Availability

Not applicable.

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
