# Peer review of "Design of a Variable Stiffness Gecko-Inspired Foot and Adhesion Performance Test on Flexible Surface"

_biomimetics, 2022, doi:10.3390/biomimetics7030125_

Round 1
Reviewer 1 Report
This contribution is in two parts: the first describes a peeling model, which allows for discreet adhesive elements and some elasticity of the surface to which the flexible peeling element is fixed. The bending stiffness of the adhering ‘tape’ can be adjusted by varying the rotational stiffness of the hinges between the units. The authors refer to joints J1 to J4 but these are not shown explicitly on Fig.2. The results from the modelling are shown in Figs 5-7 as plots of the normal component of the peel force versus the tangential component. These plots would be much more useful if the scales on each axes were the same so that the lengths of the net peel force F and the peel angle theta were true. It would also be useful for comparison with other peel models if these were plotted as peel force per unit width, i.e. N/m so that the value at theta=90* provides a useful measure of the peel energy. What part does surface energy play in the model – it would seem that the criterion for detachment is the maximum extension of the adhesion unit – but what controls this – surely the conditions at the interface between the adhesive unit tip and the counterface play a part.
The same comment on scaling goes for Fig 9, which displays the experimental results from the rather neat set-up that replicates the peel mechanics of a gecko toe. But here when theta=90 degrees peel force F would seem to be about 0.5N so if the adhesive material is 10mm wide then G= 50N/m. How does that compare with the numerical model of discrete elements each depending on van der Waal adhesion at the mushrooms. On the face of it this seems to give G=5 N/m. In these experiments were the vertical movement of the fixture and the horizontal movement of the work surface such that the peel angle was maintained constant?
The final set of tests on the performance of the gecko-like robot is impressive. But it wasn’t clear whether its performance could be predicted from the previous experimental work. The device was moving on a thin polyimide film rather than the PVC surface of the test rig. How comparable are these surface energies? It wasn’t clear to me how the first part of the paper informed the predicted performance of the gecko device.
Reviewer 2 Report
The manuscript presents an extremely interesting robot that can move on a flexible surface by attachment/detachment via dry adhesion. The results are very interesting, and worth publishing. On the other hand, there are some points that are not clear or needed more discussion, so I put some comments below.
1) Although the authors introduced two models of gecko adhesion mechanisms (Kendall tape model and PZ mode), several models to discuss the asymmetric detachment mechanism of gecko adhesion have been proposed. We can easily find them by internet search. Therefore, an introduction of previously proposed asymmetric detachment models and a discussion of the difference from this study should be added in sections 2.1 or 2.2.
2) Although rotating joints J1-J4, stiffness k and k_s, and other parameters were explained in the sentences, it becomes clearer if their positions are shown in the figure.
3) The title of section 2 is "Materials and Methods" but 2.2 and later included results and discussion. This section was more like designing processes. Therefore, please consider changing the section title.
4) "is50 mm" at line 170 ==> Need space. "is 50 mm"
